# Galaxy Groups as the Ultimate Probe of AGN Feedback

**Dominique Eckert [1],\*** , **Fabio Gastaldello [2]** , **Ewan O'Sullivan [3]** , **Alexis Finoguenov [4]** , **Marisa Brienza [5,6]**
and the X-GAP Collaboration [†]

1. Department of Astronomy, University of Geneva, Ch. d'Ecogia 16, CH-1290 Versoix, Switzerland
2. INAF—IASF Milano, Via A. Corti 12, I-20133 Milano, Italy; fabio.gastaldello@inaf.it
3. Center for Astrophysics | Harvard & Smithsonian, 60 Garden Street, Cambridge, MA 02138, USA; eosullivan@cfa.harvard.edu
4. Department of Physics, University of Helsinki, Gustaf Hällströmin katu 2, FI-00014 Helsinki, Finland; alexis.finoguenov@helsinki.fi
5. INAF—Osservatorio di Astrofisica e Scienza dello Spazio di Bologna, via Gobetti 93/3, I-40129 Bologna, Italy; marisa.brienza@inaf.it
6. Dipartimento di Fisica e Astronomia, Università di Bologna, Via P. Gobetti 93/2, I-40129 Bologna, Italy
* Correspondence: dominique.eckert@unige.ch
† https://www.astro.unige.ch/xgap/

**Abstract:** The co-evolution between supermassive black holes and their environment is most directly traced by the hot atmospheres of dark matter halos. The cooling of the hot atmosphere supplies the central regions with fresh gas, igniting active galactic nuclei (AGN) with long duty cycles. The outflows from the central engine tightly couple with the surrounding gaseous medium and provide the dominant heating source, preventing runaway cooling. Every major modern hydrodynamical simulation suite now includes a prescription for AGN feedback to reproduce the realistic populations of galaxies. However, the mechanisms governing the feeding/feedback cycle between the central black holes and their surrounding galaxies and halos are still poorly understood. Galaxy groups are uniquely suited to constrain the mechanisms governing the cooling–heating balance, as the energy supplied by the central AGN can exceed the gravitational binding energy of halo gas particles. Here, we provide a brief overview of our knowledge of the impact of AGN on the hot atmospheres of galaxy groups, with a specific focus on the thermodynamic profiles of the groups. We then present our on-going efforts to improve on the implementation of AGN feedback in galaxy evolution models by providing precise measurements of the properties of galaxy groups. We introduce the XMM-*Newton* Group AGN Project (X-GAP), a large program on XMM-*Newton* targeting a sample of 49 galaxy groups out to $R_{500c}$.

**Keywords:** black holes; galaxy groups; elliptical galaxies; intragroup medium/plasma; active nuclei; X-ray observations; hydrodynamical and cosmological simulations





## 1. Introduction

The overarching goal of galaxy evolution models is to reproduce as closely as possible the properties of the baryonic content of the Universe and its evolution. In the past decade, feedback from active galactic nuclei (AGN), i.e., the self-regulated feedback cycle between central supermassive black holes (SMBH) and their host galaxies and halos, has emerged as the most likely solution to a wide range of issues in galaxy evolution [1,2]. For instance, AGN feedback is necessary to reproduce the cut-off in the galaxy stellar mass function [3], explain the origin of the scaling relations between SMBH mass and galaxy properties [4,5], interpret the co-evolution between star formation rate and SMBH activity [6], and quench catastrophic cooling flows [7]. Modern hydrodynamical galaxy evolution models, such as EAGLE [8], BAHAMAS [9], and IllustrisTNG [10], all include a prescription for AGN feedback. The implemented feedback model ranges from pure thermal feedback to mechanical, directional feedback, e.g., [11]. While the inclusion of AGN

feedback into hydrodynamical simulations allowed, for the first time, the reproduction of a wide range of properties of the galaxy populations, the choice of the feedback scheme in state-of-the-art hydrodynamical simulations vastly differs from one simulation to the other. The parameters of the feedback model can then be tuned to reproduce a set of observables, which, in most cases, is the galaxy stellar mass function.

As stated above, the properties of the galaxy population are not sufficient on their own to create a unified AGN feedback model, as the parameters of multiple feedback models can be tuned to match the galaxy stellar mass function equally well. On the other hand, the hot gas content of galaxy groups, i.e., halos in the range $10^{13} < M_{500c} < 10^{14} M_{\odot}$, is highly sensitive to the implemented feedback scheme. At the current epoch, galaxy groups represent the peak of the halo mass density. They occupy a key regime in the evolution of galaxies, as typical $L_\star$ galaxies exist within groups of 5–20 members rather than within isolated halos [12]. In terms of the sensitivity to AGN feedback, galaxy groups occupy a transitional regime between isolated galaxies and massive galaxy clusters, as the total feedback energy is comparable to the gravitational binding energy of the gas. Their gravitational potential well is strong enough to retain a substantial hot gaseous atmosphere (the intragroup medium, hereafter, IGrM), whereas the outflows generated by the central SMBH are energetic enough to produce clearly discernible effects in the surrounding medium. Deep observations of nearby galaxy groups such as NGC 5813 [13] and NGC 5044 [14], reveal a wealth of feedback-induced features in the IGrM. Bubbles of outflowing material associated with successive outbursts of the central SMBH expand into the surrounding medium, producing pairs of cavities in the hot gas distribution [15]. The supersonic nature of the ejecta also induces shock waves propagating through the medium perpendicular to the main direction of the outflow [16]. These phenomena inject a large amount of energy into the IGrM, which offsets the radiative losses and prevents the gas from cooling. The low cooling rate eventually stops the supply of molecular gas to the central galaxy and quenches star formation [7].

In a recent paper (Eckert et al. [17]), we provided a detailed review of the AGN feedback processes in the specific context of galaxy groups. This paper provides a summary of an invited review talk given at the "AGN on the beach" conference, which took place in Tropea, Italy, from 10 to 15 September 2023. We also introduce the XMM-*Newton* Group AGN Project (X-GAP), a newly approved large program on XMM-*Newton* that aims at measuring the impact of AGN feedback on the hot atmospheres of galaxy groups in a carefully selected sample of 49 groups.

## 2. Galaxy Groups as Probes of AGN Feedback

The hot gaseous atmospheres of galaxy groups constitute a privileged environment for the study of AGN-induced feedback processes. To illustrate this point, in the left-hand panel of Figure 1 we show a composite image of the galaxy IC 2476. IC 2476 is a massive ($\log M_\star = 11.39 M_\odot$), quenched ($\log SFR = -2.124 M_\odot/\text{yr}$) elliptical galaxy at $z = 0.0472$ [18]. The blue background image is an SDSS *i*-band image, which highlights the position of the galaxy at the center of the image. While the galaxy lives in a rather poor optical environment, spectroscopic data tell us that it is the dominant galaxy of a group of 12 spectroscopic members [19]. X-ray observations of this system with XMM-*Newton* (red component in Figure 1) reveal the existence of an IGrM extending over several hundred kpc centered on IC2476. The gas temperature of ~1.2 keV is typical of the mass range populated by galaxy groups. On top of that, the image shows radio emission contours from the LOFAR Two-meter sky survey DR2 (LoTSS, [20]). The bright remnant radio galaxy B2 0924+30 [21,22], which extends over >100 kpc from the nucleus, is associated with IC 2476. A spectral aging analysis shows that the radio jets were active for a period of ~100 Myr and switched off ~50 Myr ago [22]. The radio data highlight the presence of radio-emitting electrons well beyond the size of the central galaxy, such that the bulk of the AGN energy is injected at large distances from the central galaxy within the IGrM. The outflows deposit

energy within the IGrM, which reheats the surrounding medium and eventually quenches star formation [7].

It is now well established that there exists a relation between radio-loud AGN and rich environments. Indeed, radio-loud AGN are ubiquitous among the central galaxies of cool-core clusters [23,24]. The fraction of radio-loud AGN increases steeply with galaxy stellar mass [25]. The most massive galaxies, corresponding to the central galaxies of the most massive clusters hosting the most massive black holes [26], all exhibit some level of radio emissions [25]. Within systems hosting detectable X-ray cavities [27], we observe a correlation between radio power and total jet power [28], with the total energy output exceeding the radiative output by ∼2 orders of magnitude. Moreover, cavity power correlates with the cooling luminosity of the X-ray emitting gas [15]. The energy input from the central AGN is at least sufficient to balance the radiated energy and quench catastrophic cooling flows. Finally, the blueshifted CO absorption lines reveal the existence of radially infalling gas clouds at the vicinity of the SMBH [29,30]. Therefore, the link between radio-loud AGN and their environment is now well established. Nonetheless, the total amount of energy injected in the surroundings by AGN feedback is still poorly known, especially at the mass range populated by galaxy groups.

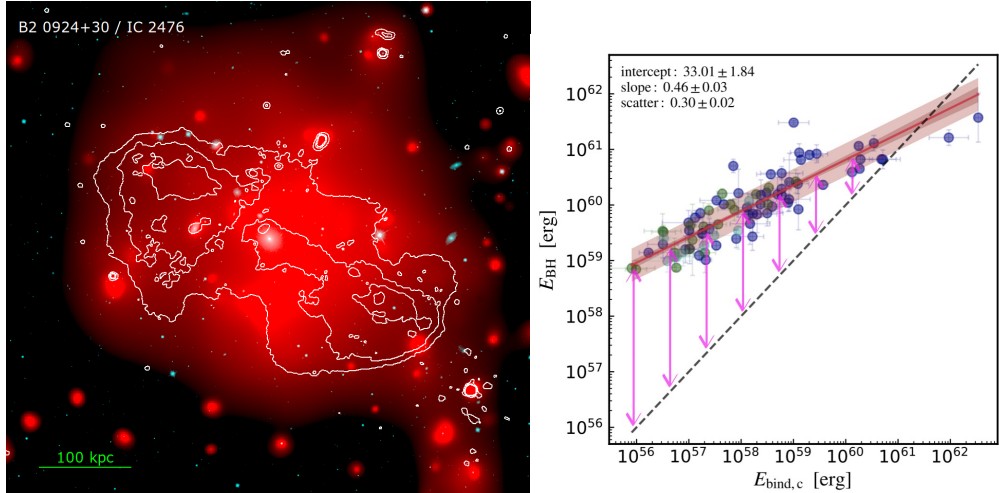

**Figure 1.** Impact of AGN feedback on galaxy group atmospheres. The left-panel shows an SDSS *i*-band image of the brightest group galaxy IC 2476 with X-ray observations of the IGrM superimposed in red. The white contours shows radio emission from the associated radio galaxy B2 0924+30 from LoTSS DR2 data [20]. The right-hand panel shows the relation between the binding energy of IGrM gas particles and the energy injection from the central BH (figure reproduced from [17]). The dashed line indicates equality between BH and binding energy.

Since the majority of the feedback energy is dissipated within the IGrM, the structural properties of the gaseous atmospheres of galaxy groups act as fossil records of the total feedback energy integrated over cosmic time. In the right-hand panel of Figure 1, we show the relation between the gas binding energy within group cores, $E_{bind} \approx 2E_{th} \propto M_{gas}k_BT$, and the available SMBH mechanical energy, $E_{BH} = \epsilon_M M_{BH}c^2$, for a sample of galaxy groups with dynamically measured BH masses [31]. The $\epsilon_M$ parameter, which represents the fraction of the accreted energy that is converted into heat, was assumed to be constant at the value of $\epsilon_M = 10^{-3}$ [32]. We can see that for low-mass (i.e., low-temperature) systems, the available BH energy largely exceeds the binding energy of gas particles in halo cores, such that the energy supplied by AGN feedback is sufficient to unbind gas particles and eject them from the halo. Therefore, the total baryon fraction of galaxy groups within $R_{500c}$ falls short of the cosmic baryon fraction $\Omega_b/\Omega_m$ (e.g., [33–36]). While most studies agree on the existence of some level of gas ejection outside of halos, the exact baryon fraction of galaxy groups is still widely debated, and the scatter in the gas fraction at a fixed mass is unknown.

Given the large amount of injected energy, we expect the effect of AGN feedback to extend throughout the entire volume of these systems, and possibly even beyond their virial radius [37–39]. In Figure 2, we show the thermodynamic profiles of simulated galaxy groups extracted from four state-of-the-art codes (EAGLE, Illustris-TNG, SIMBA, and RO-MULUS [40]). These simulations all implement a different prescription for AGN feedback. For instance, EAGLE and ROMULUS implement some version of the Booth and Schaye [41] model, where the energy deposition by AGN is fully thermal and isotropic. Conversely, SIMBA and Illustris-TNG implement dual models including a mechanical component to consider the impact of AGN jets and winds. The predicted properties of the IGrM largely differ from one simulation to the other. Models with strong feedback, such as SIMBA [42], predict substantially lower gas densities (Figure 2a) and higher entropies (Figure 2b) than models with relatively weak feedback (e.g., ROMULUS [43]). The discrepancies are echoed in the overall gas and baryon fractions within $R_{500c}$ (see the discussion in Section 5.2 of [17]): while in the galaxy cluster regime ($M_{500} > 10^{14} M_{\odot}$), the aforementioned codes predict very similar integrated gas fractions, in the group regime, the predictions differ by up to an order of magnitude. In case the implemented feedback is very strong (e.g., Illustris [44]), most of the gas is evacuated from the halo and the measured gas fractions are very low. Conversely, models implementing a more gentle feedback scheme such as EAGLE [8] predict a high gas fraction within the halo. It is worth noting that these two simulation sets predict very similar galaxy stellar mass functions, galaxy morphologies, and star formation rate distributions. Therefore, modern simulation suites have little predictive power on the baryon content of groups, even when the properties of the galaxy population are accurately reproduced.

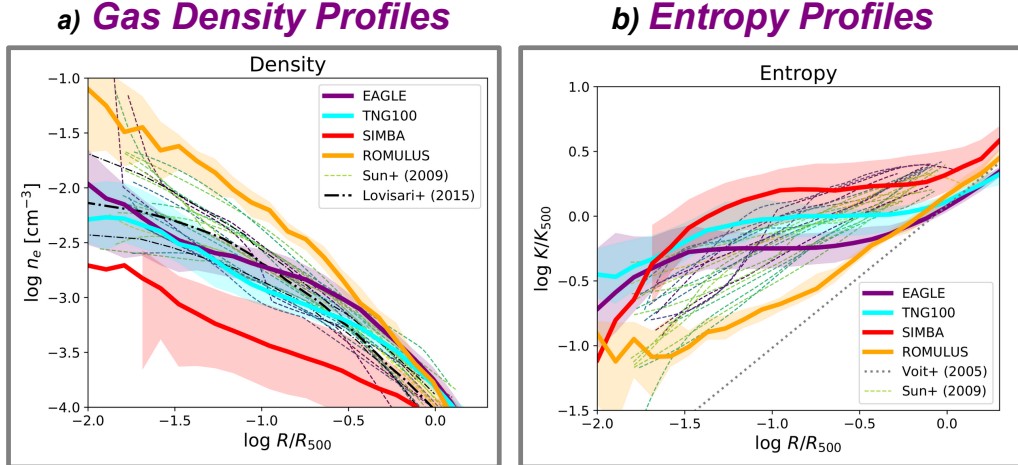

**Figure 2.** Impact of AGN feedback on the thermodynamic profiles of galaxy groups. The figure shows the profiles of IGrM gas density (**a**) and entropy (**b**) in four state-of-the-art simulations (EAGLE, Illustris-TNG100, SIMBA, and ROMULUS; [40]). Increasing the feedback energy leads to lower gas densities (**left**) and higher gas entropies (**right**), highlighting the sensitivity of the IGrM to the feedback scheme. For comparison, the dashed curves show observed galaxy group thermodynamic profiles from *Chandra* [45] and XMM-*Newton* [46].

On top of that, recent studies have shown that AGN feedback modifies the large-scale matter distribution in the Universe in a complicated way, which represents a leading source of systematic uncertainties for upcoming cosmology experiments. Baryonic processes affect the predicted matter power spectrum at the level of 10–20%, i.e., about an order of magnitude larger than the statistical precision of upcoming cosmic shear measurements [47]. The impact of baryonic processes depends very sensitively on the details of the chosen feedback model [48]. Calibrating the simulations against unbiased measurements of group gas fractions will be crucial for robustly predicting the impact of baryons on cosmological

observables and warrant the success of upcoming cosmic shear experiments like ESA's *Euclid* mission [49].

The use of the IGrM as a probe of AGN feedback has, up until this point, been limited by our observational knowledge of its properties, which is relatively primitive compared to our understanding of the intra-cluster medium in more massive halos. While it has been known for 20 years that the scaling relations deviate from self-similarity at galaxy group scale [50,51], little is known on the thermodynamic profiles of the group population as a whole. Previous works based on *Chandra* [45] and XMM-*Newton* [46,52] are based on archival studies and focus on the brightest, most nearby systems. In the vast majority of cases, the selected systems are very nearby and fill the *Chandra* and XMM-*Newton* field of view (FOV), such that direct constraints beyond $0.5R_{500c}$ are available only for a handful of systems. Alternatively, constraints on the gas fraction in galaxy groups have been obtained from X-ray surveys [35,36], such as the XXL survey, which covers an area of 50 square degrees with *XMM-Newton* [53]. However, such observations are very shallow, and the resulting uncertainties are large: the *mean* gas fraction can typically be determined with $\sim 20\%$ uncertainty at $10^{14} M_\odot$, and no information on the intrinsic scatter can be obtained. For a more comprehensive review of our knowledge of feedback effects on the IGrM, we refer the reader to Section 3 of Eckert et al. (2021) [17].

## 3. The XMM-Newton Group AGN Project (X-GAP)

To address the science questions highlighted in Section 2, we[1] initiated a program aiming at measuring the properties of the IGrM out to $R_{500c}$ in a carefully selected galaxy group sample spanning the mass range $10^{13} \lesssim M_{500c} \lesssim 10^{14} M_\odot$. However, the question of selecting a pure and unbiased sample of galaxy groups is a complex one.

Historically, samples of galaxy groups have been selected mostly based on their optical or X-ray properties. Given the low richness of galaxy groups, selection algorithms based on photometric data (e.g., redMaPPer, [54]) are strongly affected by projection effects in the mass range of galaxy groups and are thus not well suited for their detection. Conversely, large spectroscopic surveys allow for the detection of groups in three-dimensional space using algorithms such as Friends-of-Friends (FoF). Such algorithms were applied to large spectroscopic surveys (SDSS e.g., [19,55], GAMA [12]). While spectroscopic group catalogues are much cleaner than their photometric equivalent, they can still be affected by projection effects. For instance, testing their algorithm on mock data, Robotham et al. [12] estimate that 20–30% of the groups detected with at least five group members are not bona fide group-scale halos, but are rather unvirialised systems composed of several smaller halos (see also [56]). Conversely, group selection based on X-ray surveys such as the ROSAT all-sky survey (RASS, [50,57]) or eR,OSITA [58], yields group samples that are very pure, since the selection is based on the presence of a virialised IGrM. However, given the limited sensitivity of these surveys, the detection is limited to the local Universe ($z < 0.1$). On top of that, the structural properties of the IGrM, and, in particular, the presence or absence of a cool core, strongly affect the detectability of groups [59,60], such that X-ray group samples are likely biased towards the most relaxed, X-ray brightest groups.

Given our goal of selecting a highly pure and unbiased sample, following Damsted et al. [56] we attempted to combine the best features of both selection methods by cross-correlating galaxy groups selected from optical spectroscopic surveys with the presence of faint, extended X-ray sources. Starting from the SDSS FoF group catalogue of Tempel et al. [19], we selected galaxy groups with a minimum of eight spectroscopic members, and cross-correlated their position with diffuse sources selected from the RASS data. Our X-ray source detection algorithm [61] removes the central flux of X-ray point-like sources and then performs a wavelet search for core-excised extended sources on scales greater than 12′. As a result, our approach is sensitive only to large-scale X-ray emissions and is not biased toward centrally peaked systems. This strategy maximises the purity of the sample and is more complete than a pure X-ray selection. Moreover, the cross-correlation with SDSS groups provides a wealth of supporting optical data,

in particular, redshifts, velocity dispersions, star formation rates, and stellar masses. For more details on the selection of the parent sample, we refer the reader to Damsted et al. [56].

In Figure 3, we show the X-ray luminosity of the detected systems and their mass estimated from a luminosity–mass relation. The points are color-coded by the apparent size $\theta_{500}$ of $R_{500c}$. For the purpose of this study, we perform a cut according to the following criteria:

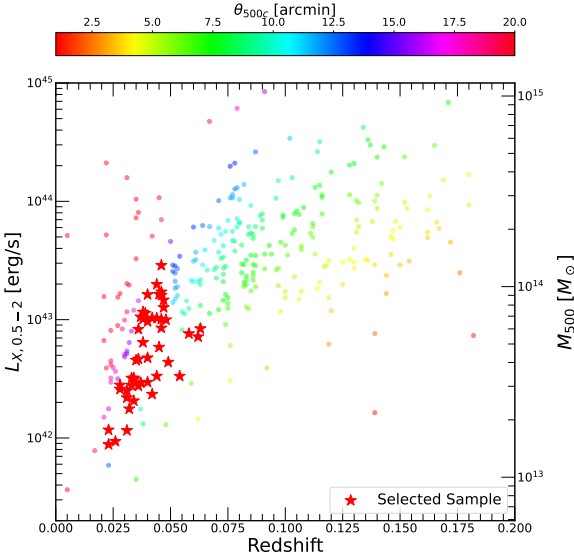

**Figure 3.** X-GAP group selection cross-matching extended sources detected in RASS with SDSS spectroscopic galaxy groups [56]. The plot shows the RASS X-ray luminosity and the corresponding mass estimated through a luminosity–mass relation as a function of the source redshift. The points are color-coded by the apparent size $\theta_{500}$ of $R_{500c}$. The selected sample is highlighted by the red stars.

- $\theta_{500} < 15'$: This ensures that $R_{500c}$ of the selected system is contained within the XMM-*Newton* FOV;
- $z < 0.06$: Given the sensitivity curve of our sample, the systems located at greater redshifts are almost exclusively clusters with $M_{200} > 10^{14} M_\odot$;
- Number of member galaxies >8: This criterion removes loosely bound galaxy systems.

The selected sample comprises 49 systems with an estimated mass $10^{13} \lesssim M_{500} \lesssim 10^{14} M_\odot$ in the narrow redshift range $0.025 < z < 0.06$. The properties and size of the sample are sufficient to investigate the dependence of the thermodynamic profiles and the gas fraction on halo mass. The sample size of 49 systems will also allow us to reliably determine the intrinsic scatter of the quantities of interest at a fixed mass. The availability of SDSS optical data and 2MASS and WISE near-IR data ensures that we will be able to estimate the stellar fractions as well, allowing us to determine the total baryon fraction and the relative contribution of gas and stars. Finally, the vast majority of the selected systems fall within the LoTSS footprint, which will allow us to correlate the properties of the IGrM with those of the central radio galaxies.

To demonstrate the capabilities of our observing strategy, we searched the XMM-*Newton* archive and analysed the existing public data for a subset of four groups that were already observed by XMM-*Newton* and that are representative of the systems available in the archive in terms of their mass and redshift. We reduced the available data using `XMMSASv19.1` and extracted thermodynamic profiles and hydrostatic mass profiles using the Python package `hydromass` [62]. The resulting thermodynamic and mass profiles are shown in Figure 4. The determined masses lie in the range $5 \times 10^{13} M_\odot < M_{500} < 1.5 \times 10^{14} M_\odot$, and the recovered profiles extend to $R_{500c}$, which shows that our selection criteria are adequate. This analysis shows that we are able to determine $f_{\text{gas}}$ inside $R_{500c}$ *without extrapolation*. The availability of velocity dispersion measurements for all the systems

provides an additional, independent estimate of the group's mass, which is important to verify that our mass measurements are not severely biased by the hydrostatic assumption.

Interestingly, all four systems show very flat surface brightness profiles and a strong deficit of gas within their central regions. The corresponding entropy profiles show a striking entropy excess extending all the way out to $R_{500c}$ and high central cooling times. The selected systems also have low pressure and do not show any central temperature drop. Our selection process may thus unveil a population of low surface brightness groups that was missed by the standard RASS detection pipelines and is absent from previous galaxy group samples [45,46]. If confirmed, this finding will have profound consequences for AGN feedback models (see Figure 2), as it would imply that the properties of the IGrM are more diverse than previously thought.

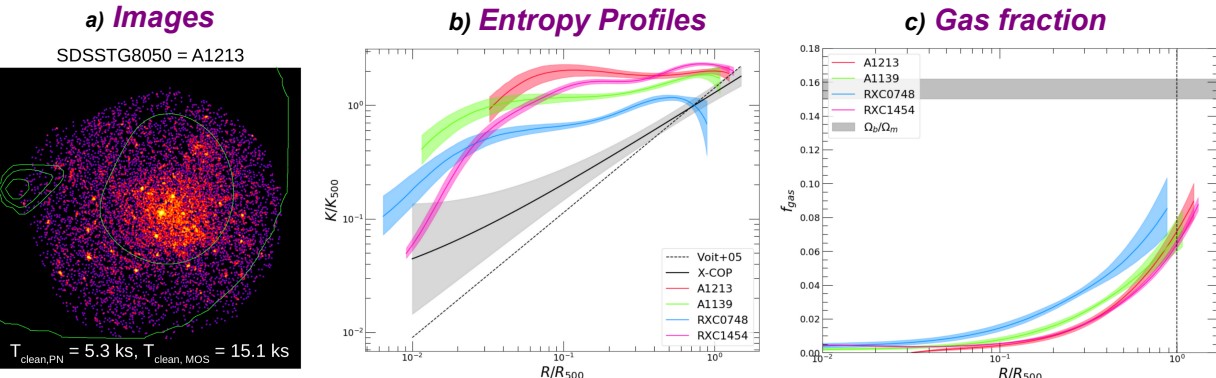

**Figure 4.** Analysis results for four groups matching the X-GAP selection criteria with available observations in the XMM-*Newton* archive. The selected groups exhibit a flat brightness distribution, as highlighted in the (**a**), where we show the combined XMM-*Newton*/EPIC image of one of the systems (SDSSTG 8050 = A1213) with RASS contours overlaid in green. The low surface brightness implies a strong entropy excess (**b**) extending all the way out to $R_{500c}$. The gas is evacuated towards the outskirts, where the gas fraction rises sharply (**c**). Given our selection process, we are able to determine the gas fraction at $R_{500c}$ without requiring any extrapolation.

The X-GAP program aims at providing observations of similar quality over the entire sample of 49 groups presented in Figure 3. The program was accepted into A priority during XMM-*Newton* AO-19 for a total observing time of 852ks, which constitutes the largest observing program awarded that year. The X-GAP program will yield at least 20,000 source photons for each target, which is similar to the four archival observations presented in Figure 4. We note, however, that the observations previously available in the archive are skewed towards the bright end of the full sample, such that the full sample covers a substantially wider mass and luminosity range. In Appendix A, we provide a master table with the description of the 49 groups selected for X-ray follow-up with XMM-*Newton*. Out of the 49 groups, 11 were already observed previously, whereas the remaining 38 are new observations. We also provide an image gallery of all the observed systems. All the observations were reduced using XMMSASv20.0 and the X-COP analysis pipeline [63]. The images provided in the gallery are adaptively smoothed, background subtracted, and vignetting corrected maps in the [0.7–1.2] keV band. The location of SDSS FoF member galaxies is also highlighted with the cyan squares. The very high success rate of our observing strategy is apparent in Figures A1–A4. Indeed, the vast majority of the selected groups host bright diffuse X-ray emission extending over 10 arcmin or more, which confirms the efficiency of our selection process. The data quality is sufficient to obtain results of similar quality as those presented in Figure 4 for a large sample of galaxy groups. Follow-up campaigns in the radio and optical are now being undertaken, which will eventually bring us a more comprehensive view of the properties of galaxy groups.

## 4. Conclusions

In this paper, we gave an overview of an invited talk on the impact of AGN on the hot atmospheres of galaxy groups given at the "AGN on the beach" conference in Tropea, Italy. The paper can be summarised in the following way:

- Galaxy groups occupy a key mass regime where the energy injected by AGN feedback is sufficient to affect the baryonic properties of the system over the entire volume, yet not so strong that most of the baryons are evacuated from the halo. The hot atmospheres of galaxy groups can be used as a fossil record of the feedback energy dissipated by AGN over the entire history of these systems;
- The mechanical energy injected by radio AGN is deposited far outside of the central galaxy into the surrounding IGrM (see the left-hand panel of Figure 1), which is eventually responsible for cutting the supply of fresh gas on the central galaxy and quenching star formation;
- The properties of the IGrM predicted by modern hydrodynamical simulations implementing different feedback models strongly vary from one simulation to another (see Figure 2). Calibrating the feedback model against high-fidelity measurements of IGrM properties is therefore a key step toward creating a unified model of energy injection by AGN in galaxy evolution models;
- To advance our understanding of IGrM properties, we were recently awarded the XMM-*Newton* Group AGN Project (X-GAP), a large program on XMM-*Newton* targeting a sample of 49 galaxy groups selected by cross-matching SDSS FoF catalogues with weak RASS extended sources (see Figure 3; [56]). In Appendix A, we provide a master table describing the selected sources and an XMM-*Newton* image gallery.

**Author Contributions:** D.E.: lead author and X-GAP PI. F.G. and E.O.: X-GAP co-PIs. A.F.: X-GAP sample selection. M.B.: X-GAP LOFAR images. Conceptualization: all, Methodology: D.E. and A.F., Software: D.E., Validation: D.E., Formal Analysis: D.E., A.F. and M.B., Investigation: E.O. and F.G., Resources: all, Data curation: D.E., Writing—original draft: D.E., Writing—review & editing: D.E., Visualization: D.E., Supervision: D.E., Project Administration: D.E., E.O. and F.G., Funding acquisition: all. All authors have read and agreed to the published version of the manuscript.

**Funding:** D.E. is supported by the Swiss National Science Foundation (SNSF) through grant agreement 200021_212576. M.B. acknowledges support from the agreement ASI-INAF n. 2017-14-H.O and from the PRIN MIUR 2017PH3WAT "Blackout".

**Data Availability Statement:** X-GAP data will be publicly released on the project website: https://www.astro.unige.ch/xgap/ (accessed on 5 May 2024).

**Acknowledgments:** D.E. thanks Ilaria Ruffa for the organisation of a beautiful conference and the SOC for the kind invitation.

**Conflicts of Interest:** The authors declare no conflicts of interest.

## Appendix A. X-GAP Master Table and Image Gallery

**Table A1.** Master table of X-GAP galaxy groups.

| ID | RA (deg) | Dec (deg) | z | $F_{[0.1-2.4]}$ ($10^{-13}$ ergs/cm$^2$/s) | $N_{gal}$ | $\sigma_v$ (km/s) | $M_{200c}$ ($10^{13} M_{\odot}$) | BGG Name | BGG z |
|---|---|---|---|---|---|---|---|---|---|
| 828 | 153.41 | −0.93 | 0.046 | 54.22 ± 4.70 | 55 | 750 | 19.6 ± 1.8 | UGC 5515 | 0.0453 |
| 885 | 117.05 | 18.55 | 0.047 | 26.43 ± 3.84 | 62 | 626 | 2.9 ± 0.6 | MCG+03-20-013 | 0.0225 |
| 1011 | 223.60 | 16.36 | 0.046 | 42.15 ± 6.30 | 11 | 374 | 2.4 ± 0.5 | IC 4516 | 0.0241 |
| 1162 | 232.37 | 7.57 | 0.044 | 41.03 ± 7.68 | 18 | 316 | 4.9 ± 0.7 | NGC 5931 | 0.0267 |
| 1398 | 228.20 | 7.43 | 0.046 | 30.26 ± 4.54 | 74 | 617 | 4.3 ± 0.8 | UGC 9767 | 0.0279 |
| 1601 | 161.09 | 14.08 | 0.034 | 7.24 ± 3.39 | 23 | 355 | 5.3 ± 1.0 | NGC 3357 | 0.0323 |

**Table A1.** *Cont.*

| ID | RA (deg) | Dec (deg) | z | $F_{[0.1-2.4]}$ $(10^{-13}$ ergs/cm$^2$/s) | $N_{gal}$ | $\sigma_v$ (km/s) | $M_{200c}$ $(10^{13} M_\odot)$ | BGG Name | BGG z |
|---|---|---|---|---|---|---|---|---|---|
| 1695 | 155.42 | 23.92 | 0.04 | 40.96 ± 4.25 | 44 | 441 | 4.0 ± 1.1 | NGC 3216 | 0.0326 |
| 2424 | 241.61 | 15.69 | 0.04 | 7.44 ± 2.70 | 85 | 627 | 5.3 ± 0.9 | MCG+03-41-123 | 0.0335 |
| 2473 | 174.80 | 55.67 | 0.063 | 8.24 ± 2.30 | 19 | 368 | 5.5 ± 0.8 | MCG+09-19-143 | 0.0335 |
| 2620 | 233.13 | 4.68 | 0.039 | 29.99 ± 4.77 | 43 | 498 | 6.9 ± 0.9 | UGC 9886 | 0.0351 |
| 3128 | 243.15 | 29.48 | 0.032 | 7.00 ± 1.73 | 27 | 313 | 10.1 ± 1.0 | NGC 6086 | 0.0352 |
| 3460 | 170.74 | 34.11 | 0.044 | 21.28 ± 4.31 | 26 | 359 | 5.0 ± 0.7 | UGC 6394 | 0.0358 |
| 3513 | 252.57 | 23.58 | 0.036 | 8.59 ± 3.13 | 24 | 362 | 11.7 ± 0.8 | NGC 6233 | 0.0361 |
| 3669 | 162.19 | 22.22 | 0.048 | 17.21 ± 3.22 | 30 | 426 | 12.5 ± 1.1 | MCG+04-26-010 | 0.0379 |
| 4047 | 174.01 | 55.08 | 0.058 | 8.88 ± 2.42 | 35 | 413 | 12.5 ± 1.2 | MCG+09-19-131 | 0.0389 |
| 4436 | 159.30 | 50.12 | 0.046 | 16.02 ± 2.61 | 30 | 562 | 11.0 ± 1.0 | NGC 3298 | 0.0392 |
| 4654 | 188.92 | 26.52 | 0.023 | 6.87 ± 2.36 | 24 | 297 | 7.0 ± 1.1 | NGC 4555 | 0.0390 |
| 4936 | 145.89 | 39.42 | 0.042 | 5.33 ± 1.72 | 18 | 442 | 15.7 ± 1.0 | UGC 5193 | 0.0392 |
| 5742 | 176.59 | 33.16 | 0.034 | 11.29 ± 2.65 | 42 | 425 | 5.2 ± 1.1 | NGC 3880 | 0.0402 |
| 6058 | 156.09 | 41.71 | 0.045 | 11.54 ± 2.63 | 17 | 403 | 11.5 ± 1.1 | MCG+07-22-001 | 0.0413 |
| 6159 | 203.99 | 33.43 | 0.026 | 5.69 ± 1.81 | 23 | 321 | 11.4 ± 1.4 | IC 4305 | 0.0427 |
| 8050 | 169.09 | 29.25 | 0.046 | 32.34 ± 5.90 | 68 | 582 | 17.7 ± 2.1 | MCG+05-27-037 | 0.0437 |
| 8102 | 238.76 | 41.58 | 0.033 | 10.37 ± 3.61 | 26 | 492 | 15.7 ± 1.4 | MCG+07-33-011 | 0.0444 |
| 9178 | 162.50 | 0.32 | 0.04 | 11.97 ± 3.12 | 18 | 281 | 8.1 ± 1.1 | MCG+00-28-017 | 0.0445 |
| 9370 | 196.24 | 43.55 | 0.038 | 17.94 ± 2.63 | 29 | 340 | 21.9 ± 1.2 | MCG+07-27-026 | 0.0457 |
| 9399 | 140.85 | 22.31 | 0.035 | 15.02 ± 2.88 | 35 | 561 | 10.1 ± 1.0 | UGC 4991 | 0.0451 |
| 9647 | 138.41 | 29.99 | 0.023 | 9.12 ± 3.31 | 29 | 347 | 11.3 ± 1.1 | NGC 2783 | 0.0452 |
| 9695 | 165.24 | 10.51 | 0.038 | 32.36 ± 4.54 | 47 | 577 | 16.1 ± 1.8 | NGC 3492 | 0.0453 |
| 9771 | 205.60 | 29.82 | 0.044 | 6.89 ± 2.19 | 33 | 482 | 13.2 ± 1.3 | NGC 5275 | 0.0461 |
| 10094 | 151.72 | 14.37 | 0.031 | 10.61 ± 2.70 | 35 | 364 | 14.6 ± 1.3 | NGC 3121 | 0.0468 |
| 10159 | 206.32 | 23.22 | 0.031 | 9.28 ± 2.71 | 31 | 406 | 11.2 ± 1.3 | LEDA 48750 | 0.0466 |
| 10842 | 164.55 | 1.60 | 0.04 | 24.23 ± 3.49 | 51 | 439 | 6.8 ± 0.8 | UGC 6057 | 0.0340 |
| 11320 | 146.72 | 54.45 | 0.045 | 32.34 ± 4.54 | 45 | 492 | 15.5 ± 1.5 | MCG+09-16-044 | 0.0458 |
| 11631 | 239.59 | 18.08 | 0.046 | 19.02 ± 3.00 | 41 | 504 | 2.8 ± 0.5 | 2MASX J15582067+1804512 | 0.0580 |
| 11844 | 216.17 | 26.63 | 0.038 | 28.27 ± 4.44 | 9 | 304 | 2.4 ± 0.5 | MCG+05-34-033 | 0.0222 |
| 12349 | 200.06 | 33.14 | 0.037 | 31.14 ± 3.41 | 41 | 406 | 5.2 ± 1.1 | NGC 5098 | 0.0336 |
| 15354 | 181.10 | 42.56 | 0.054 | 4.50 ± 1.96 | 13 | 372 | 5.8 ± 1.1 | 2MASX J12042469+4233432 | 0.0427 |
| 15641 | 141.97 | 29.99 | 0.028 | 14.63 ± 3.20 | 12 | 258 | 6.9 ± 1.1 | IC 2476 | 0.0472 |
| 15776 | 164.43 | 37.65 | 0.036 | 14.51 ± 3.01 | 14 | 291 | 4.5 ± 0.9 | MCG+06-24-039 | 0.0405 |
| 16150 | 123.65 | 55.14 | 0.033 | 11.97 ± 3.60 | 24 | 346 | 5.0 ± 1.1 | MCG+09-14-020 | 0.0361 |
| 16386 | 249.32 | 44.42 | 0.031 | 4.91 ± 1.43 | 10 | 267 | 8.9 ± 0.8 | 2MASX J16370588+4416111 | 0.0389 |
| 16393 | 152.71 | 54.21 | 0.047 | 22.84 ± 3.59 | 46 | 295 | 4.8 ± 0.7 | MCG+09-17-036 | 0.0298 |
| 21128 | 197.18 | 13.81 | 0.062 | 7.27 ± 2.43 | 13 | 376 | 4.8 ± 0.6 | 2MASX J13084384+1348248 | 0.0265 |
| 22635 | 230.05 | 25.72 | 0.034 | 10.93 ± 3.10 | 30 | 438 | 3.8 ± 0.6 | MCG+04-36-038 | 0.0325 |
| 28674 | 203.24 | 32.61 | 0.037 | 8.66 ± 1.85 | 20 | 282 | 11.9 ± 1.2 | MCG+06-30-029 | 0.0371 |
| 35976 | 136.98 | 49.60 | 0.036 | 25.88 ± 4.33 | 31 | 391 | 9.3 ± 1.6 | MCG+08-17-034 | 0.0571 |
| 39344 | 184.91 | 28.50 | 0.028 | 13.51 ± 2.73 | 21 | 270 | 9.9 ± 1.7 | LEDA 39736 | 0.0612 |
| 40241 | 239.17 | 20.17 | 0.049 | 7.23 ± 1.92 | 34 | 439 | 5.7 ± 1.5 | 2MASX J15564131+2010172 | 0.0555 |
| 46701 | 123.19 | 54.14 | 0.042 | 23.45 ± 3.74 | 17 | 369 | 9.4 ± 1.9 | 2MASX J08124599+5408228 | 0.0607 |

Column description: 1: Group ID [1]. 2: Right ascension [1]. 3: Declination [1]. 4: ROSAT all-sky survey flux in the [0.1–2.4] keV band [2]. 5: Number of FoF galaxies with spectroscopic redshift [1]. 6: Gapper velocity dispersion in km/s [2]. 7: Halo mass within an overdensity of 200 times the critical density estimated from a mass-luminosity relation [2]. 8: Name of brightest group galaxy [3]. 9: Redshift of brightest group galaxy [3]. References: [1] [19]; [2] [56]; [3] This work.

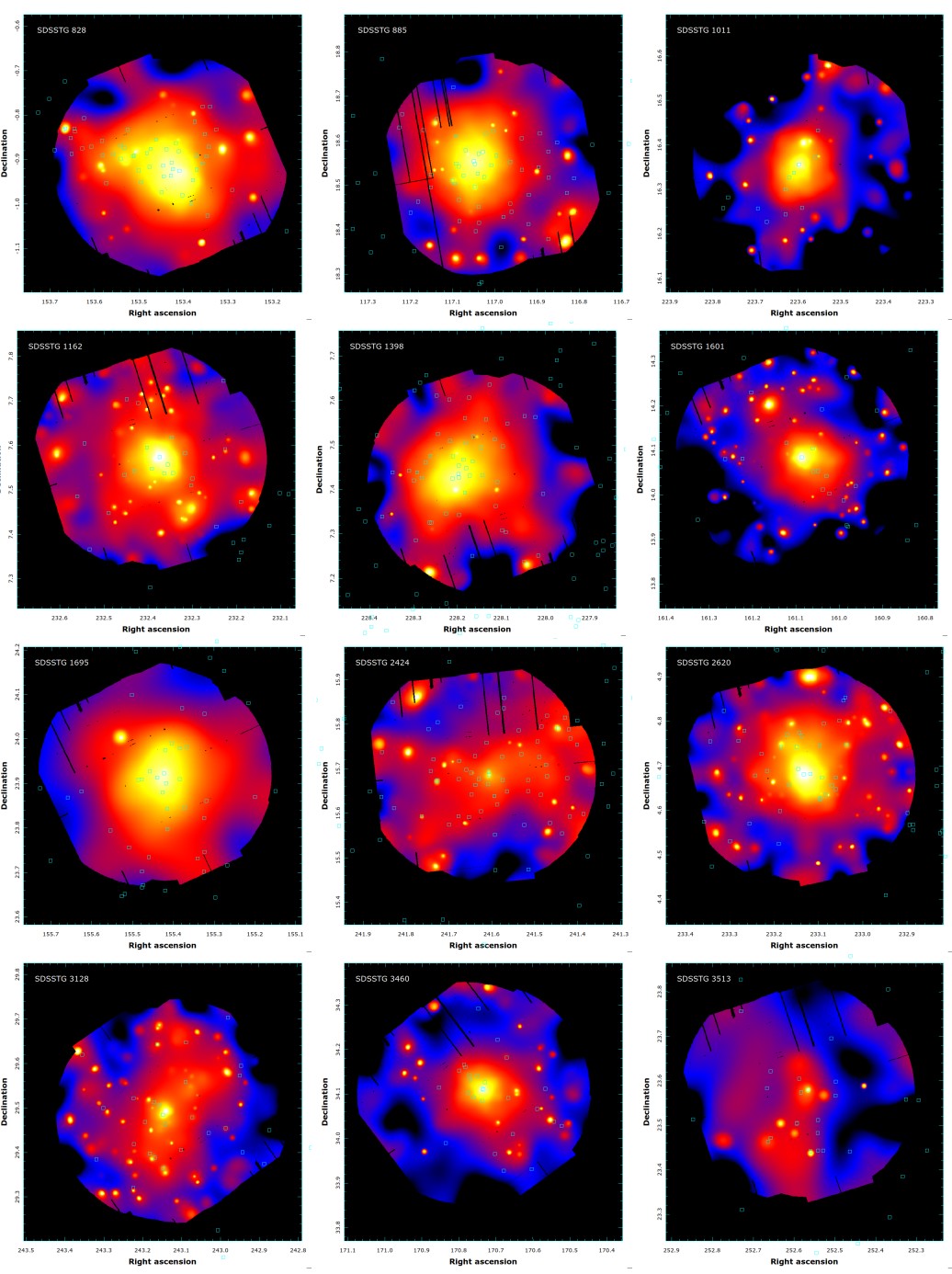

**Figure A1.** Background subtracted, vignetting corrected, and adaptively smoothed *XMM-Newton*/EPIC maps of X-GAP groups in the [0.7–1.2] keV band. Shown are SDSSTG 828, SDSSTG 885, SDSSTG 1011 (top row), SDSSTG 1162, SDSSTG 1398, SDSSTG 1601 (second row), SDSSTG 1695, SDSSTG 2424, SDSSTG 2620 (third row), SDSSTG 3128, SDSSTG 3460, SDSSTG 3513 (bottom row). The magenta squares show the position of SDSS member galaxies selected using the FoF algorithm [19]. In all cases, the axes are right ascension and declination in unit of degrees.

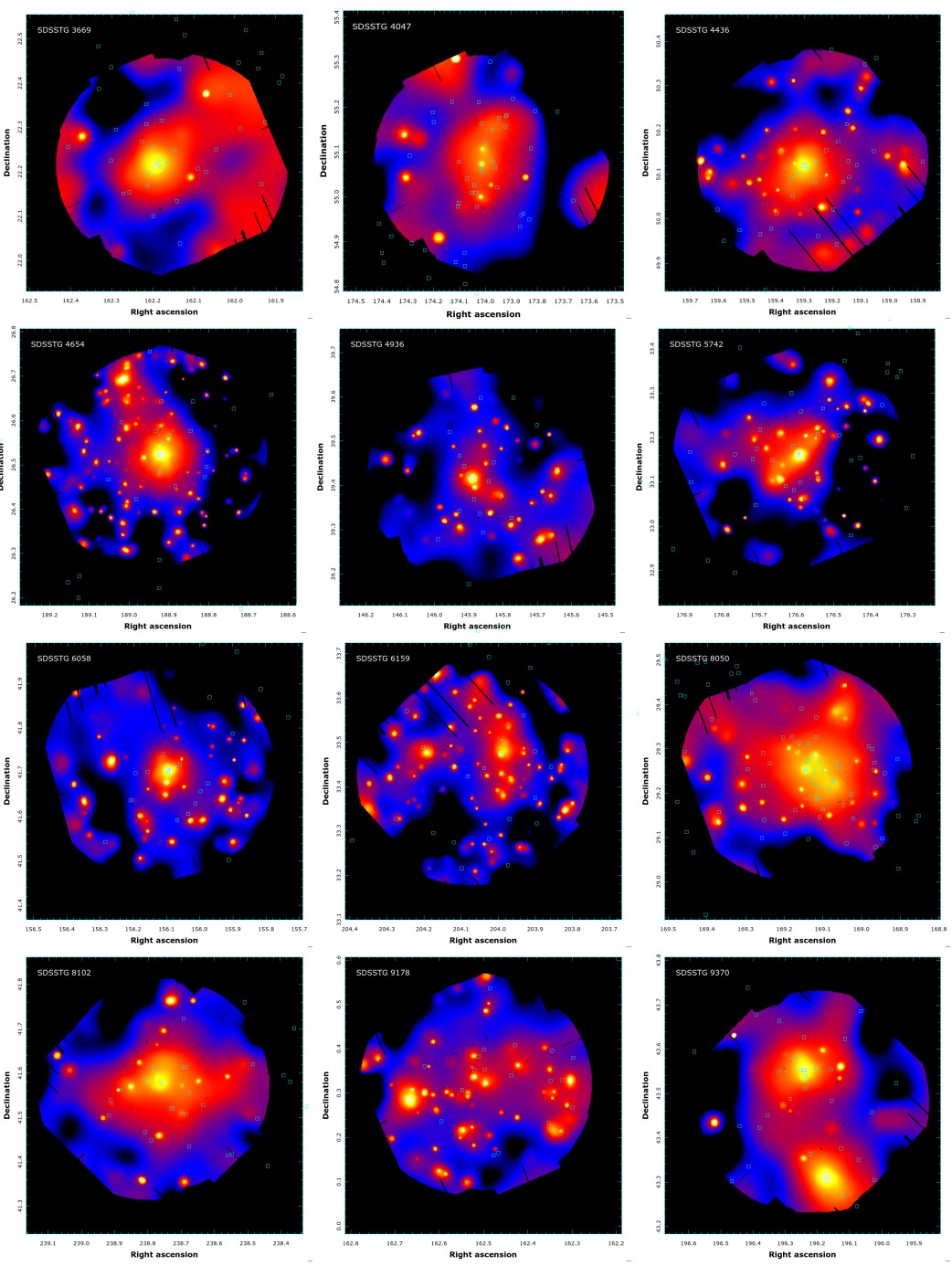

**Figure A2.** Same as Figure A1 for SDSSTG 3669, SDSSTG 4047, SDSSTG 4436 (top row), SDSSTG 4654, SDSSTG 4936, SDSSTG 5742 (second row), SDSSTG 6058, SDSSTG 6159, SDSSTG 8050 (third row), SDSSTG 8102, SDSSTG 9178, SDSSTG 9370 (bottom row).

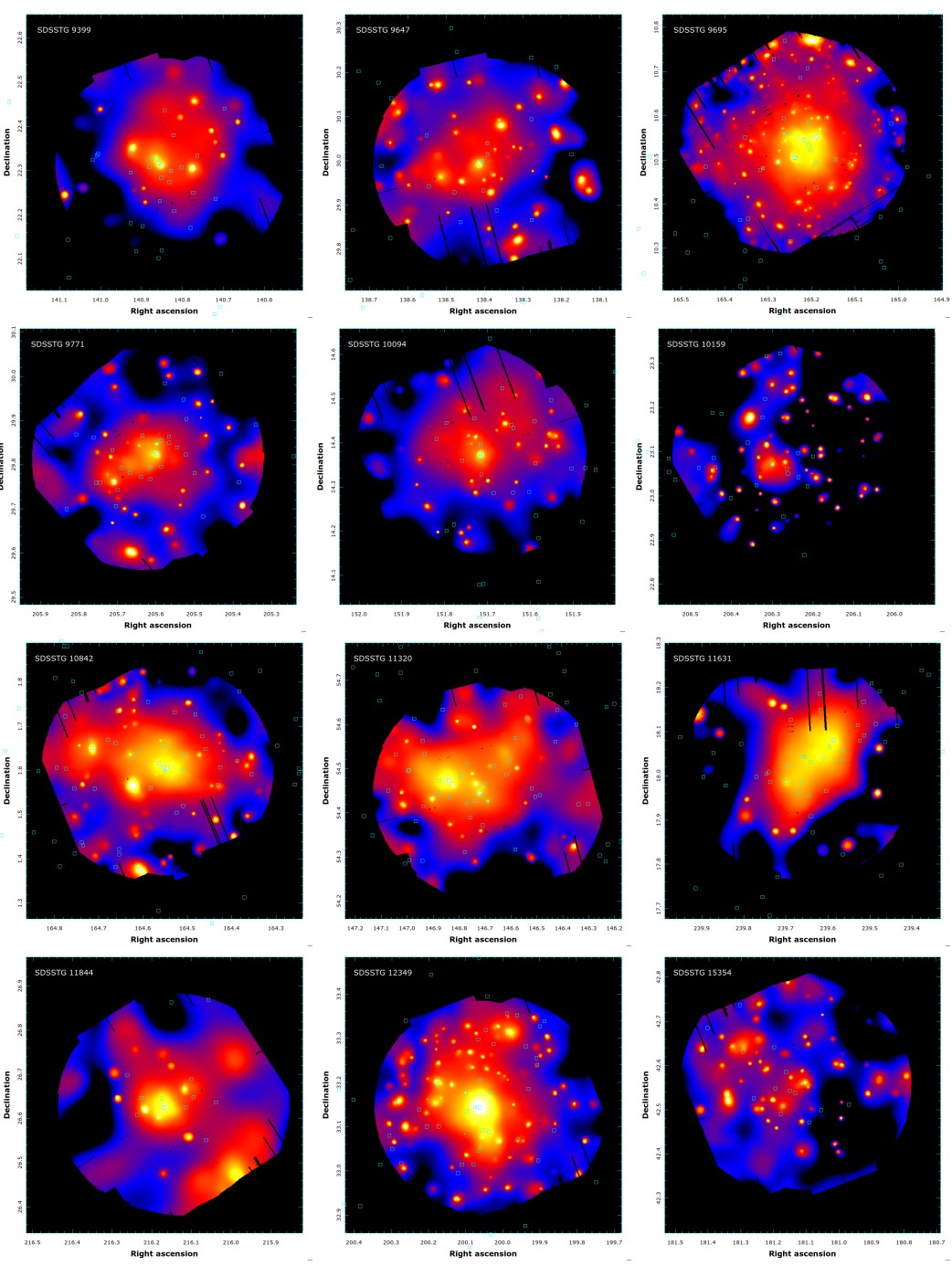

**Figure A3.** Same as Figure A1 for SDSSTG 9399, SDSSTG 9647, SDSSTG 9695 (top row), SDSSTG 9771, SDSSTG 10094, SDSSTG 10159 (second row), SDSSTG 10842, SDSSTG 11320, SDSSTG 11631 (third row), SDSSTG 11844, SDSSTG 12349, SDSSTG 15354 (bottom row).

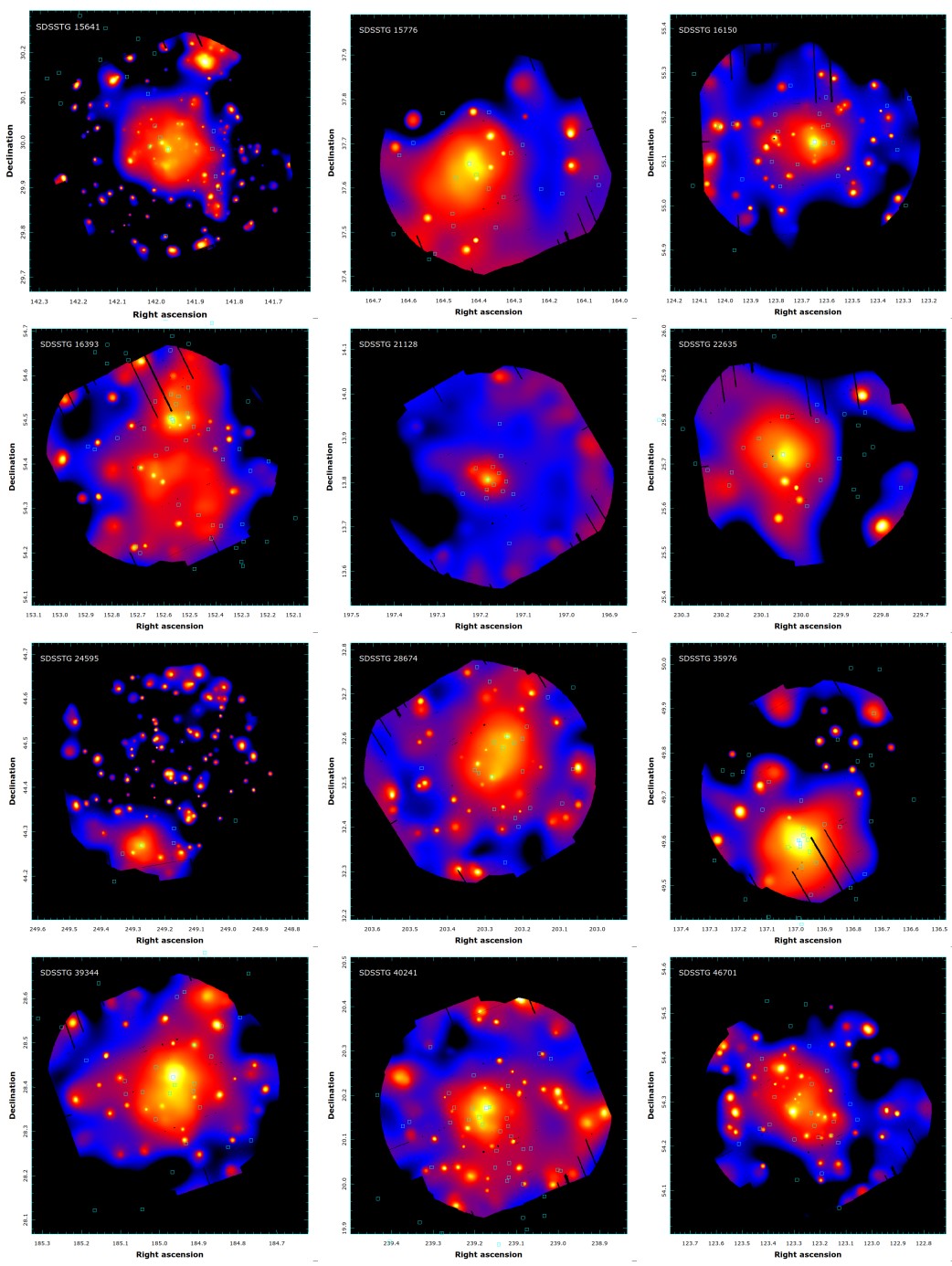

**Figure A4.** Same as Figure A1 for SDSSTG 15641, SDSSTG 15776, SDSSTG 16150 (top row), SDSSTG 16393, SDSSTG 21128, SDSSTG 22635 (second row), SDSSTG 24595, SDSSTG 28674, SDSSTG 35976 (third row), SDSSTG 39344, SDSSTG 40241, SDSSTG 46701 (bottom row).

## Note

1   https://www.astro.unige.ch/xgap/blog/people, accessed on 5 May 2024.

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
