# Peer review of "Galaxy Groups as the Ultimate Probe of AGN Feedback"

_galaxies, doi:10.3390/galaxies12030024_

Round 1

Reviewer 1 Report

Comments and Suggestions for Authors

I agree that AGN feedback in groups is a key piece of the puzzle for how black holes affect and are affected by their host galaxies and large scale environments. The basic premise that the hot atmosphere cools in clusters to trigger AGN is, however, not supported observationally.  In fact, the opposite is true. We observe FRI and FRII jets in cluster environments coming from the central galaxy in low excitation regime. Whatever the reasons for the different jet morphology (engine versus environment), we would expect random distributions of such objects. Instead, we find that the black hole mass of the low excitation FRII radio galaxies are smaller than the black hole masses of the low excitation FRI radio galaxies. Jet power increases with black hole mass in models of jets from black holes. So why are the more powerful jets coming from the smaller black holes? In addition to this, one would expect a random distribution in jet morphology over time. Instead, we find the FRII to be at slightly higher redshift. If we include the high excitation FRII radio galaxies, we see a further shift in the average redshift to higher values (see Macconi, Torresi, Grandi et al 2020). Hence, your basic premise behind this work is flawed and needs revisiting. The most likely explanation appears to be some kind of time evolution between the families of radio galaxies. I suggest you recast your idea of using groups to understand AGN feedback in a better context.

Author Response

We certainly agree with the referee that the distribution of FRI vs FRII radio galaxies is puzzling and that the mechanism responsible for the difference in jetted vs non-jetted AGN is unknown. However, we still think the relation between hot halos and AGN, if not unique, is indisputable. In the following we provide several reasons supporting our claim.

  • We have known since the early 1990s (e.g. Burns et al. 1990) that radio-loud AGN are ubiquitous among the BCGs of cool-core clusters. More recent works (e.g. Sabater et al. 2019) have shown that the fraction of radio-loud AGN increases steeply with stellar mass, and that the most massive galaxies, corresponding to the BCGs of the most massive clusters hosting the most massive black holes, all exhibit some level of radio emission.
  • Considering the energetics, we know that the kinetic output of FRIs largely exceeds their radiative output. There is a well-known relation between radio power and total jet power in systems hosting cavities (Cavagnolo et al. 2010), and cavity power is known to correlate with the cooling luminosity (Birzan et al. 2008), such that there is definitely some relation between hot halo cooling and radio AGN triggering.
  • Observations of red- or blue-shifted CO absorption features by ALMA (Tremblay et al. 2016; Oosterloo et al. 2024) reveal the existence of radially infalling gas clouds at the vicinity of SMBH, which further strengthens the probable link between SMBH feeding and AGN cooling
  • Recently, several works (Bogdan et al. 2018; Gaspari et al. 2019; Lakchaura et al. 2019) found that the properties of the hot atmospheres correlate with SMBH mass. In particular, the relation between gas temperature and BH mass was found to be tighter than the corresponding galaxy scaling relations, in particular the relation between BH mass and K-band luminosity.

Given the limited scope of this particular paper, which aims at summarizing the content of a talk and presenting a recent XMM large program, it is not possible for us to review all the existing phenomenology within our manuscript. However, we have added a new paragraph in Sect. 2 to describe the above response within the manuscript.

Reviewer 2 Report

Comments and Suggestions for Authors

Report on "Galaxy Groups as  the Ultimate Probe of AGN Feedback" by Eckert et al.

This article is a combination of a review and an accepted proposal which is likely to yield important results in near future. Therefore its content is well within the scope of Galaxies Journal.

The article is well motivated. The introduction to sample selection and example of how the unambiguous results might be arrived at has been written lucidly.

Some minor comments I have which might add a little value to it. (a) In Figure 1 (right) a line should be written on the dashed curve in the text and in the Caption. (b) In texts relating to Figure 2, at a sentence should be spent to explain how the four models, namely, EAGLE, Illustris-TNG, SIMBA, and ROMULUS differ from each other, and how the physical inputs vary which caused so much variation of the density of entropy profile.

At the end the authors add some speculations on what the proposal is expected to yield. That would clearly establish their intuitive power, once such results are indeed observed.

I recommend the publication after these changes are made.

Author Response

We thank you for your comments on the manuscript. We have revised the paper accordingly. The changes with respect to the previous version are highlighted in boldface.

  • In the caption of Fig. 1 we have added a description of the dashed line, which represents the equality line
  • In Sect. 2 we added a couple of sentences describing the differences between the various feedback models. While the feedback model in EAGLE and ROMULUS is purely thermal, Illustris-TNG and SIMBA implement a hybrid model where a fraction of the energy output is mechanical.

Reviewer 3 Report

Comments and Suggestions for Authors

This paper explores the timely topic of AGN feedback, with a focus on the important regime of galaxy groups. This is an important topic in the current literature and relevant with up-to-date issues being discussed. The paper is limited in scope, providing a brief motivation for the topic of studying AGN feedback in groups, before using this to introduce a new survey X-GAP. It does not intend/attempt to give a comprehensive review. Given the limited intended scope of the publication, this brief summary paper is worthy of publication. I have only a few minor suggestions:

Abstract:

-              I didn’t really understand what was meant by “benchmarks on the properties of galaxy groups”. I suppose the point here was this new survey will provide tight observational constraints on gas distributions in groups (and scatter within the population), which can then be used to test different theoretical AGN feedback prescriptions. Perhaps some slight re-wording would make this clearer.

Section 1:

-              Line 51: I think it is important to clarify what these “pre-defined observables” are. This is because there are several more subtle observables, for which there is not so good agreement between different models (e.g., galaxy sizes, sSFR distributions, AGN fractions etc.). I believe the point made here is that the stellar mass function (and maybe black hole mass – stellar mass relations) can be reproduced using different methods.

-              Line 68 was a bit ambiguous. The first time I read it, it sounded like the energy *prevented* star-formation quenching. A slight re-phrase of the sentence would reduce this ambiguity.

Section 2

-              Line 90: Please specific how the image “clearly shows that the bulk of the AGN energy is injected at large distances”. Presumably the depression in X-rays, but this might not be obvious to all readers.

-              Line 97: I suppose this is some convention, that can not be easily changed, but could perhaps be clarified in this work…The efficiency parameter is multiplied by the black hole mass and it is stated that this is the fraction of the total accreted matter that is converted to heat. However, the black hole mass is material that has not, and will not, be converted to energy. Some fraction of the total accreted matter will have been converted to energy, whilst the rest will have been added to the black hole as mass. Anyway, I think this is just a confusing convention that could be more clearly defined.

-              Line 122: again, being specific what is mean by “very similar galaxy populations” would be better.

-              Line 145: A brief explanation of what XXL is, would be appropriate.

-              In this section (when discussing Figure 2) it would seem appropriate to mention briefly what was found in other recent work looking at gas expulsion in haloes from cosmological simulations and how they relate to different feedback models, e.g.,

https://ui.adsabs.harvard.edu/abs/2019MNRAS.485.3783D
https://ui.adsabs.harvard.edu/abs/2020MNRAS.491.4462D/abstract

https://ui.adsabs.harvard.edu/abs/2024ApJ...960...28V/abstract

Section 3

-              Line 208. It looks like 11 groups were already observed, (from line 233), so I was left wondering why these 4 were selected as examples and if these are biased in any way of the total 49 groups. Some comment on how representative they are would be appropriate (e.g., did they have some special reason that they have archival observations; were they the brightest?).

Figures:

-              The axis labels in both figure 4 and appendix figures are very hard to see (they are too small). I would consider a better way to present these so they can be read. For the appendix figures, perhaps a scale bar would be a simple fix?

Author Response

We thank you for the detailed review of our manuscript. We took your points into account within the revised version. The changes with respect to the previous version are highlighted in boldface in the manuscript.

  • Abstract: we replaced the term "benchmarks" with "measurements"
  • Sect. 1: instead of "pre-defined observables" we now refer specifically to the galaxy stellar mass function
  • Sect. 1: we rephrased the sentence on feedback-induced quenching
  • Sect. 2: here we meant to say that since the radio source is much larger than the size of the central galaxy, most of the energy is deposited at much larger radii than the limit of the stellar halo. We have rephrased the corresponding sentence.
  • Sect. 2: while we obviously agree that the BH mass itself cannot be converted into energy, the available mechanical energy considered here refers to the typical energy that an AGN would inject into its environment, which given the Eddington limit depends on the BH mass. Therefore, as shown in Gaspari & Sadowski (2017), if the BH is accreting primarily from the cooling halo then the BH energy is proportional to the BH mass.
  • Sect. 2: we specified the various galaxy properties.
  • Sect. 2: we introduced a brief description of the XXL survey.
  • Sect. 2: the recommended citations on gas ejection were included.
  • Sect. 3: the four groups that we considered initially are not representative of X-GAP as a whole but their masses and redshifts are representative of the 11 archival systems. We added a couple of sentences in the text to clarify this point. Our aim with these figures was to showcase the capabilities of X-GAP rather than making a strong scientific point, thus for the purpose of this paper the exact choice of the systems is not very important.
  • We have increased the size of the labels in Fig. 4. The labels of the figures in the appendix are simply R.A. and Dec, which we indicated within the caption.

Round 2

Reviewer 1 Report

Comments and Suggestions for Authors

I accept the revisions